# Slow-Binding Inhibition of Tyrosinase by *Ecklonia cava* Phlorotannins

**DOI:** 10.3390/md17060359

**Published:** 2019-06-16

**Authors:** Jang Hoon Kim, Sunggun Lee, Saerom Park, Ji Soo Park, Young Ho Kim, Seo Young Yang

**Affiliations:** 1College of Pharmacy, Chungnam National University, Daejeon 34134, Korea; oasis5325@gmail.com (J.H.K.); staroot99@naver.com (S.L.); tofhadlzz@naver.com (S.P.); bjs442@naver.com (J.S.P.); 2Biospectrum Life Science Institute, A-1805, U-TOWER, 767, Sinsu-ro, Suji-gu, Yongin-si, Gyeonggi-do 16827, Korea

**Keywords:** *Ecklonia cava*, Laminareaceae, phlorotannins, competitive inhibitor, slow binding inhibitor

## Abstract

Tyrosinase inhibitors improve skin whitening by inhibiting the formation of melanin precursors in the skin. The inhibitory activity of seven phlorotannins (**1**–**7**), triphlorethol A (**1**), eckol (**2**), 2-phloroeckol (**3**), phlorofucofuroeckol A (**4**), 2-*O*-(2,4,6-trihydroxyphenyl)-6,6′-bieckol (**5**), 6,8′-bieckol (**6**), and 8,8′-bieckol (**7**), from *Ecklonia cava* was tested against tyrosinase, which converts tyrosine into dihydroxyphenylalanine. Compounds **3** and **5** had IC_50_ values of 7.0 ± 0.2 and 8.8 ± 0.1 μM, respectively, in competitive mode, with *K*_i_ values of 8.2 ± 1.1 and 5.8 ± 0.8 μM. Both compounds showed the characteristics of slow-binding inhibitors over the time course of the enzyme reaction. Compound **3** had a single-step binding mechanism and compound **5** a two-step-binding mechanism. With stable AutoDock scores of −6.59 and −6.68 kcal/mol, respectively, compounds **3** and **5** both interacted with His85 and Asn260 at the active site.

## 1. Introduction

Melanin is produced in skin epidermal cells to protect the skin from ultraviolet radiation [1]. The overproduction and accumulation of melanin causes age spots, freckles, melisma, and hyperpigmentation [1,2]. Tyrosinase (EC 1.14.18.1), which belongs to the type 3 copper protein family, is a key enzyme in melanogenesis [1,3]. It is a multifunctional oxide that contains a copper in each of two sets of three histidine residues in the active site. [4]. Furthermore, it catalyzes the hydroxylation of l-tyrosine to l-3,4-dihydroxyphenylalanine (l-DOPA), and the subsequent oxidation of l-DOPA to DOPA quinone [3,5]. These products are used in melanin biosynthesis [4]. Recently, the tyrosinase inhibitors kojic acid and arbutin were developed to improve skin whitening and as anti-hyperpigmentation agents [3,6]. However, they had undesirable side effects, including dermatitis, skin irritation, and DNA damage [1,4]. Much research has examined a variety of natural plants, with the aim of developing new inhibitors [7]. This study also sought alternative inhibitors without adverse effects.

*Ecklonia cava*, in the family Laminariaceae, grows along the coast of Jeju Island, Korea [8] and is widely used in Korea as a food ingredient, animal feed, and medicine [9]. *E. cava* contains phlorotannins in polymerized phloroglucinol units [8]. Phlorotannins are secondary metabolites that are important to human health [10]. These compounds have antioxidant, anti-cancer, anti-allergy, and anti-HIV bioactivities [10,11]. In addition, the compounds eckol, phlorofucofureckol A, and dieckol, from the edible brown alga *Ecklonia*
*stolonifera*, exhibit anti-tyrosinase activity at macromolar concentrations [12]. 

This study evaluated the tyrosinase inhibitory activity of minor components from *E. cava*. Among seven isolated compounds (**1**–**7**), two minor compounds (**3** and **5**) had IC_50_ values of less than 10 μM, and also exhibited competitive and slow-binding inhibition of tyrosinase. This study also shows how these compounds interact with the catalytic site of tyrosinase via molecular docking.

## 2. Results and Discussion

### 2.1. Isolation and Identification

The ethyl acetate fraction of ethanol extract of *E. cava* was known to suppress the catalytic reaction of tyrosinase [13]. Phlorotannins, eckol, triphlorethol-A, phlorofucofuroeckol A, and dieckol, were reported to be isolated from this fraction [14,15]. To isolate phlorotannins which have not been identified as tyrosinase inhibitors, dried *E. cava* powder (1.1 kg) was extracted by reflux with 80% ethanol for 72 h. The concentrated (80%) ethanol extract (290.0 g) was progressively divided into *n*-hexane, ethyl acetate, and butanol layers. The ethyl acetate layer (54.9 g) was subjected to open column chromatography with silica gel and C-18 resin to obtain seven phlorotannins (**1**–**7**). By comparing spectroscopic data with reported results, the compounds were identified as triphlorethol A (**1**) [16], eckol (**2**) [8], 2-phloroeckol (**3**) [16], phlorofucofuroeckol A (**4**) [12], 2-*O*-(2,4,6-trihydroxyphenyl)-6,6′-bieckol (**5**) [17], 6,8′-bieckol (**6**) [18], and 8,8′-bieckol (**7**) [8] (Figure 1).

### 2.2. Inhibition of Phlorotannins on Tyrosinase

Phlorotannins of *E. cava* and *E. stolonifera* were reported to have the inhibitory activity on mushroom tyrosinase [12,19]. 7-phloroeckol and dieckol were revealed to be the potential inhibitors within micromole concentration [13,19]. This study evaluated the ability of the isolated phlorotannins **1**–**7** to suppress the catalytic reaction of tyrosinase over time, in the absence or presence of inhibitor. Their inhibitory activity was calculated using equation (1). A commercial tyrosinase inhibitor was used as a positive control (kojic acid; IC_50_ = 25.0 ± 0.4 μM). To identify potent inhibitors, the inhibitory activity of all of the isolated compounds at 100 μM was tested against tyrosinase in vitro (Table 1). Of these, compounds **2**–**5** were confirmed to have inhibitory activity exceeding 50%. Serial dilutions were used to calculate the IC_50_ values. The tyrosinase inhibitory activity increased in a dose-dependent fashion (Figure 2A). Compounds **2**–**5** showed inhibitory activity, with IC_50_ values of 7.0 ± 0.2 to 66.4 ± 0.1 μM (Table 1). Of these, compounds **3** and **5** had inhibitory activity at < 10 μM. Interestingly, the structure of compound **5** contained the moiety of compound **3**. 

Our studies and reported results showed that phlorotannins, 2-phloroeckol, 7-phloroeckol dieckol, and 2-*O*-(2,4,6-trihydroxyphenyl)-6,6′-bieckol, substituted with phloroglucinol in eckol increased the inhibitory effect on tyrosinase compared to other phlorotannins [12,19]. Recently, secondary metabolites from terrestrial plants have been studied primarily for tyrosinase inhibitors [4,5,6,7]. Morin, norartocarpetin, artogomezianone, and (-)-*N*-formylanonaine have been revealed as the potential inhibitors [4,5,6,7]. Interest in the components from seaweed led to the study of the isolation of phlorotannin. It has been proven that they improve skin whitening [12,13,19]. These results suggest that 2-phloroeckol and 2-*O*-(2,4,6-trihydroxyphenyl)-6,6′-bieckol are tyrosinase inhibitors that can replace phenolic compounds [4,5,6,7].

To gain insight into the interaction of the enzyme with phlorotannins, as tyrosinase inhibitors, the products of the catalytic reaction were determined by ultraviolet-visible photometry after the inhibitor was added to the enzyme solution. The IC_50_ values of compounds **3** and **5** increased linearly with [S]/Km (Figure 2B). Furthermore, compounds **3** and **5** were produced with a set of the family liners by various inhibitors at substrate concentrations ranging from 0.156 to 2.50 mM. As shown in Lineweaver–Burk plots (Figure 2C,D), they had different values of −1/Km and the same 1/Vmax. The interactions with tyrosinase were competitive, with respective *K*_i_ values of 8.2 ± 1.1 and 5.8 ± 0.8 μM based on Dixon plots (Figure 2E,F, Table 1). Phlorotannins from seaweeds have been used to develop the non-competitive tyrosinase inhibitors 7-phloroeckol, dieckol, and phlorofucofuroeckol A, and the competitive inhibitors phloroglucinol and eckstolonol [12,13]. We found that compounds **3** and **5** competitively inhibited the catalysis of tyrosinase and might be useful for improving whitening.

### 2.3. Slow-Binding Inhibition

To confirm time-dependent inhibition by the potential inhibitors (compounds **3** and **5**), the substrate was added into a mixture of ligand (3.1 μM) that was preincubated with tyrosinase. Over time, their inhibitory activities increased. To calculate the slow-binding parameters (*k*_3_, *k*_4_, *k*_5_, *k*_6_, and *k*_app_
*i*), the progress curves were analyzed using equation (2), with increasing concentrations of compounds **3** and **5** (Figure 3C,D); a replot of *k*_obs_ was obtained as [*I*] (Figure 3E,F). The replot of compound **3** was a straight line that fit equation (3). When forming an encounter complex (EI) of the receptor with the ligand, compound **3** slowly binds to the active site of tyrosinase according to slow-binding mechanism A (a single-step binding mechanism) (Figure 3E) [20]. In comparison, the replot of *k*_obs_ for compound **5** fit a hyperbolic equation (5) based on mechanism B (a two-step binding mechanism). This indicates that enzyme isomerization (E^*^I) results in slow bonding after the ligand rapidly interacts with the receptor [20,21]. Table 2 shows the kinetic parameters for the time-dependent inhibition of tyrosinase by the inhibitors. The two strongest inhibitors had different slow-binding mechanisms. Inhibitor **5**, which has a higher molecular weight, induced a new conformational state of the enzyme.

### 2.4. Molecular Docking

Molecular docking was performed to determine how the ligand interacts with the active site of tyrosinase. The best fit of their complex was considered to have the lowest energy, as calculated using Autodock 4.2. As indicated in Figure 4A–E and Table 3, the AutoDock scores of compounds **3** and **5** showed a trend similar to their IC_50_ values in vitro. They covered the hole and area around the active site, with calculated energies of −6.59 and −6.68 kcal/mol, respectively (Figure 4A). Compound **3** was held by six hydrogen bonds with three amino acids (Glu256 2.82 Å, Asn260 2.73 Å, and Met280 2.68 Å) and hydrophobic interactions with 15 amino acids (Figure 4B,C, Table 3). The phloroglucinol moiety of compound **3** had a π–π bond interaction with His85 at 9.4Å distance (Figure 4E). For compound **5**, seven hydroxyl groups were close to eight amino acids: Lys79 (2.95Å), Asn81 (2.77 and 2.85Å), His85 (3.01, 3.17, and 2.79Å), Cys83 (2.71Å), Asn260 (2.69Å), Gly281 (2.69Å), His244 (3.30Å), and Glu322 (2.68Å). Furthermore, there were hydrophobic interactions with 18 amino acids (Figure 4D,E, Table 3).

## 3. Materials and Methods 

### 3.1. General Experimental Procedures

NMR experiments were conducted on an ECA500 (JEOL, Tokyo, Japan), with the chemical shift referenced to the residual solvent signals, using methanol-*d*_4_ and DMSO-*d*_6_ as a solvent. TLC analysis was performed on silica-gel 60 F_254_ and RP-18 F_254S_ plates (both 0.25 mm layer thickness, Merck, Darmstadt, Germany). Compounds were visualized by dipping plates into 10% (*v*/*v*) H_2_SO_4_ reagent and then air heat treated at 300 °C for 15 s. Silica gel (Merck 60A, 70–230 or 230–400 mesh ASTM) and reversed-phase silica gel (YMC Co., ODS-A 12 nm S-150, S-75 μm) were used for open column chromatography. Tyrosinase (T3824), kojic acid (K3125) and l-tyrosine (T3754) were purchased from Sigma-Aldrich (ST. Louis, Mo, USA). UV–vis and fluorescence spectra were measured by TECAN infinite 200 PRO^®^ spectrophotometer (Zurich, Switzerland).

### 3.2. Plant Material

*E. cava* was purchased from a herbal market in Jeju Island, Korea, on May 2015. One of the author (Prof. Y.H. Kim) identified this brown algal species. A voucher specimen (CNU-15005) was deposited at the Herbarium, College of Pharmacy, Chungnam National University (CNU).

### 3.3. Extraction and Isolation

The dried powder (1.0 kg) of *E. cava* was refluxed with 80% EtOH (16 L) for 72 h, and the ethanol extract was concentrated under vacuum to yield a dark green residue (290.0 g). The residue (290.0 g) was suspended in H_2_O (2.0 L), and the aqueous layer was partitioned with *n*-hexane, ethyl acetate, and *n*-butanol. The ethyl acetate layer (54.9 g) was subjected to silica gel column eluted with CH_2_Cl_2_:MeOH (8:1→2:1) to obtain four fractions (5A–5D). Fraction 5B was chromatographed on silica gel column eluting with CHCl_3_:MeOH:H_2_O (7:1:0.1→1:1:0.1) to afford six fractions (5BA–5BF). Fraction 5BA was chromatographed on C-18 column eluting with MeOH:H_2_O (1:3→1:1) to isolate compound **1** (304.4 mg). Fraction 5BB was chromatographed on silica gel column eluting with CHCl_3_:MeOH:H_2_O (4:1:0.1→1:1:0.1) to obtain two fractions (5BBA–5BBB). Fraction 5BBB was subjected to C-18 column eluting with (CH_3_)_2_CO:MeOH:H_2_O (1:1:3→1:1:1) to isolate compound **3** (69.0 mg). Fraction 5BD was subjected to C-18 column eluted with (CH_3_)_2_CO:H_2_O (1:3→1:1) to obtain compound **2** (128.5 mg). Fraction 5BE was subjected to C-18 column using MeOH:H_2_O (1:3→1:1) to afford compound **4** (230.2 mg). The fraction 5BF was subjected to C-18 column eluted with MeOH:H_2_O (1:4→1:1) to obtain compound **5** (52.7 mg). Fraction 5C was subjected to C-18 column eluted with (CH_3_)_2_CO:H_2_O (1:5→1:1) to obtain compound **7** (110.5 mg). Fraction 5D was chromatographed on C-18 column eluting with (CH_3_)_2_CO:H_2_O (1:4→1:1) to obtain compound **6** (67.5 mg).

#### Compound **1**


White powder; ESI-MS: *m*/*z* 373.2 [M − H]^−^; ^1^H-NMR (methanol-*d*_4_, 400 MHz): δ 6.02 (1H, d, *J* = 2.8 Hz, H-5), 5.97 (2H, d, *J* = 2.1 Hz, H-6″, H-2″), 5.89 (1H, t, *J* = 2.1 Hz, H-4″), 5.86 (2H, s, H-5′, H-3′), 5.72 (1H, d, *J* = 2.8 Hz, H-3). ^13^C-NMR (methanol-*d*_4_, 100 MHz): δ 162.5 (C-1″), 160.4 (C-3″, C-5″), 156.5 (C-4′), 156.2 (C-4), 153.8 (C-6), 152.7 (C-2), 152.2 (C-6′, C-2′), 125.7 (C-1), 124.7 (C-1′), 98.0 (C-3), 97.5 (C-4″), 96.2 (C-5′, C-3′), 95.4 (C-2″, C-6″), 95.0 (C-5).

#### Compound **2**


Light brown powder; ESI-MS: *m*/*z* 371.2 [M − H]^−^; ^1^H-NMR (DMSO-*d*_6_, 300 MHz): δ 9.54 (1H, s, OH-9), 9.48 (1H, s, OH-4), 9.21 (2H, s, OH-2, OH-7), 9.17 (2H, s, OH-3′, OH-5′), 6.14 (1H, s, H-3), 5.97 (1H, d, *J* = 2.5 Hz, H-8), 5.81 (1H, d, *J* = 1.7 Hz, H-4′), 5.80 (1H, d, *J* = 2.5 Hz, H-6), 5.72 (2H, d, *J* = 1.7 Hz, H-2′, H-6′). ^13^C-NMR (DMSO-*d*_6_, 75 MHz): δ 160.4 (C-1′), 158.9 (C-3′, C-5′), 153.1 (C-7), 146.1 (C-9), 146.0 (C-2), 142.6 (C-5a), 142.0 (C-4), 137.2 (C-10a), 123.2 (C-1), 122.7 (C-9a), 122.2 (C-4a), 98.5 (C-8), 98.2 (C-3), 96.3 (C-4′), 93.9 (C-6), 93.7 (C-2′, C-6′).

#### Compound **3**


Light brown powder; ESI-MS: *m*/*z* 495.2 [M − H]^−^; ^1^H NMR (DMSO-*d*_6_, 400 MHz): δ 9.52 (1H, s, OH-9), 9.41 (1H, s, OH-4), 9.19 (1H, s, OH-7), 9.13 (2H, s, OH-3′, OH-5′), 9.08 (2H, s, OH-2″, OH-6″), 8.98 (1H, s, OH-4″), 5.98 (1H, d, *J* = 2.2 Hz, H-8), 5.87 (2H, d, *J* = 1.8 Hz, H-2′, H-6′), 5.85 (2H, s, H-3″, H-5″), 5.84 (1H, t, *J* = 1.8 Hz, H-4′), 5.83 (1H, s, H-3), 5.80 (1H, d, *J* = 2.2 Hz, H-5). ^13^C NMR (DMSO-*d*_6_, 100 MHz): δ 160.5 (C-1′), 158.9 (C-3′, C-5′), 154.9 (C-4″), 153.2 (C-7), 151.3 (C-2″, C-6″), 147.9 (C-2), 146.3 (C-9), 142.6 (C-5a), 141.7 (C-4), 137.2 (C-10a), 124.4 (C-4a), 122.7 (C-1), 122.0 (C-9a), 98.7 (C-8), 96.4 (C-4′), 96.0 (C-3), 94.9 (C-3″, C-5″), 94.1 (C-2′, C-6′), 93.9 (C-6).

#### Compound **4**


Amorphous powder; ESI-MS: m/z 601.2 [M − H]^−^; 1H-NMR (DMSO-d6, 300 MHz): δ 10.14 (1H, s, OH-14), 9.87 (1H, s, OH-4), 9.85 (1H, s, OH-10), 9.44 (1H, s, OH-2), 9.21 (2H, s, OH-3′, OH-5′), 9.18 (2H, s, OH-3″, OH-5″), 8.22 (1H, s, OH-8), 6.73 (1H, s, H-13), 6.44 (1H, s, H-9), 6.31 (1H, s, H-3), 5.84 (2H, d, *J* = 1.8 Hz, H-4′, H-4″), 5.78 (2H, d, *J* = 1.8 Hz, H-2′, H-6′), 5.73 (2H, d, *J* = 1.8 Hz, H-2″, H-6″). ^13^C-NMR (DMSO-*d*_6_, 75 MHz): δ 160.2 (C-1′), 160.0 (C-1″), 159.0 (C-3″, C-5″), 158.8 (C-3′, C-5′), 150.8 (C-12a), 150.4 (C-10), 149.5 (C-11a), 147.0 (C-2), 146.5 (C-8), 144.8 (C-14), 142.0 (C-4), 136.8 (C-15a), 134.0 (C-5a), 126.4 (C-14a), 122.6 (C-4a), 122.5 (C-1), 120.1 (C-11), 103.4 (C-7), 103.2 (C-6), 99.1 (C-9), 98.3 (C-3), 96.5 (C-4″), 96.4 (C-4′), 94.8 (C-13), 93.7 (C-2′, C-6′), 93.5 (C-2″, C-6″).

#### Compound **5**


Amorphous powder; ESI-MS: *m*/*z* 865.3 [M − H]^−^; ^1^H NMR (DMSO-*d*_6_, 400 MHz): δ 9.29 (1H, s, OH-9), 9.26 (1H, s, OH-9′), 9.15 (5H, s, OH-3″, OH-5″, OH-3‴, OH-5‴, OH-2′), 9.09 (IH, s, OH-4), 9.07 (1H, s, OH-4′), 9.05 (2H, s, OH-2⁗, OH-6⁗), 8.93 (1H, s, OH-4⁗), 8.66 (IH, s, OH-7), 8.62 (1H, s, OH-7′), 6.08 (1H, s, H-3′), 6.06 (1H, s, H-8), 6.04 (1H, s, H-8′), 5.90 (2H, s, *J* = 2.1 Hz, H-2″, H-6″), 5.83 (1H, d, *J* = 2.1 Hz, H-4″), 5.83 (2H, s, H-3‴, H-5‴), 5.80 (1H, s, H-3), 5.79 (1H, d, *J* = 3.0 Hz, H-4⁗), 5.74 (2H, d, *J* = 3.0 Hz, H-2⁗, H-6⁗). ^13^C NMR (DMSO-*d*_6_, 100 MHz): δ 160.6 (C-1″, C-1‴), 158.9 (C-3″, C-3‴, C-5″, C-5‴), 154.9 (C-4⁗), 151.5 (C-7, C-7′), 151.3 (C-2⁗, C-6⁗), 147.4 (C-2′), 145.6 (C-2), 144.7 (C-9, C-9′), 142.0 (C-4′), 141.8 (C-4), 141.5 (C-5a′), 141.5 (C-5a), 137.4 (C-10a′), 137.3 (C-10a), 124.8 (C-1⁗), 123.8 (C-1″), 122.9 (C-9a, C-9a′), 122.5 (C-1), 122.1 (C-4a, C-4a′), 99.8 (C-6′), 99.7 (C-6), 98.0 (C-8, C-8′), 97.8 (C-3′), 96.4 (C-4‴, C-4″), 96.3 (C-3), 94.9 (C-3⁗, C-5⁗), 94.2 (C-2″, C-2‴), 93.8 (C-″, C-6‴).

#### Compound **6**


Brown amorphous powder; ESI-MS: *m*/*z* 741.2 [M − H]^−^; ^1^H NMR (DMSO-*d*_6_, 400 MHz): δ 9.46 (1H, s, OH-2′), 9.28 (1H, s, OH-9′), 9.17 (3H, s, OH-3‴, OH-5‴, OH-4′), 9.12 (3H, s, OH-3″, OH-5″, OH-2), 9.07 (1H, s, OH-4), 8.76 (1H, s, OH-7′), 8.65 (1H, s, OH-7), 7.90 (1H, s, OH-9), 6.19 (1H, d, H-8), 6.10 (1H, s, H-3′), 6.06 (1H, s, H-3), 5.97 (1H, s, H-6′), 5.81 (2H, d, *J* = 2.0 Hz, H-4″, H-4‴), 5.78 (2H, d, *J* = 2.0 Hz, H-2‴, H-6‴), 5.76 (2H, d, *J* = 2.0 Hz, H-2″, H-6″). ^13^C NMR (DMSO-*d*_6_, 100 MHz): δ 160.4 (C-1″, C-1‴), 158.8 (C-3″, C-3‴, C-5″, C-5‴), 151.7 (C-7), 151.5 (C-7′), 145.8 (C-2), 145.4 (C-2′), 144.6 (C-9), 144.3 (C-9′), 141.8 (C-4), 141.6 (C-4′), 141.2 (C-5a), 140.9 (C-5a′), 137.2 (C-10a), 137.2 (C-10a′), 123.5 (C-1), 123.2 (C-1′), 123.0 (C-9a), 122.8 (C-9a′), 122.3 (C-4a), 121.8 (C-4a′), 104.1 (C-8′), 99.3 (C-6), 98.0 (C-8), 96.1 (C-4″), 96.0 (C-4‴), 93.8 (C-2‴, C-6‴), 93.7 (C-2″, C-6″), 93.6 (C-6′).

#### Compound **7**


Brown amorphous powder; ESI-MS: *m*/*z* 741.2 [M − H]^−^; ^1^H NMR (DMSO-*d*_6_, 300 MHz): δ 9.46 (1H, s, OH-4), 9.18 (1H, s, OH-2), 9.13 (2H, s, OH-3′, OH-5′), 8.77 (1H, s, OH-9), 7.91 (1H, s, OH-7), 6.18 (1H, s, H-3), 5.98 (1H, s, H-6), 5.81 (1H, t, *J* = 1.8 Hz, H-4′), 5.76 (2H, d, *J* = 1.8 Hz, H-2′, H-6′). ^13^C NMR (DMSO-*d*_6_, 75 MHz): δ 160.9 (C-1′), 159.2 (C-3′, C-5′), 152.2 (C-7), 146.3 (C-2), 145.1 (C-9), 142.3 (C-4), 141.8 (C-5a), 137.7 (C-10a), 123.8 (C-1), 123.6 (C-9a), 122.9 (C-4a), 104.7 (C-8), 98.6 (C-3), 96.7 (C-4′), 94.4 (C-2′, C-6′), 94.3 (C-6).

### 3.4. Tyrosinase Assay

To evaluate inhibitory activity on tyrosinase with isolated compounds, 130 μL of tyrosinase in 0.05 mM phosphate buffer (pH 6.8) was divided in 96 well plates [19]. 20 μL of compound concentrations ranging from 1–0.015 mM was added. 50 μL of 1.5 mM L-tyrosine in phosphate buffer was diluted into the mixture for calculating the inhibitory activity. 50 μL of 10–0.62 mM L-tyrosine in buffer was added to analyze initial velocity (*v*_0_). After starting their reaction for 20 min, an amount of the product was detected at UV–vis 475 nm. The inhibitory activity was analyzed according to Equation (1)
Inhibitory activity rate (%) = [(ΔC − ΔS)/ΔC] × 100(1)

### 3.5. Slow-Binding Inhibition Analysis

The progress curves were calculated to Equation (2) by Morrison according to time course for 420 s.
[P] = *v_s_t* + [(*v*_0_ − *v*_s_)/*k*_obs_(1 − e^−*k_obst_*^)](2)
where *t* is time, [P] is product intensity, *v*_0_ and *v*_s_ are the initial and steady-state reaction velocities, and *k*_obs_ is the apparent first-order rate concentration.
*k*_obs_ = *k*_4_ + *k*_3_[*I*](3)
(4)$$K_{i}^{app}=k_{4}/k_{3}$$

This is a linear Equation (3) for slow-binding mechanism A. $K_{i}^{app}$ is the apparent value of an inhibitor by Equation (4). Where [*I*] is inhibitor concentration.
(5)$$k_{\mathrm{obs}}=k_{6}+[(k_{5}\times [I])/(K_{i}^{app}+[I])$$

Equation (5) of hyperbola is for the calculation of rate constant in slow-binding mechanism B.

### 3.6. Molecular Docking

This simulation was performed as described previously. The 3D structure of protein coded with 2Y9X was achieved from RCSB homepage [22]. This was added hydrogen atoms and assigned with gasteiger charges by using Autodocktools. Ligands were built and minimized energy by MM2 with Chem3D Pro. They were established the grid containing activity site with number of points (X: 80, Y: 80, Z: 80). DPS file was contained to set up Lamarckian genetic algorithm for docking of the ligand into receptor (runs 50 and the maximum number was set as long). Their results were represented with Chimera (version 3.13.1, San Francisco, CA, USA) and LigPlot (version 4.5.3, Cambridge, UK). 

## 4. Conclusions

Seven phlorotannins (**1**–**7**) from *E. cava* were isolated using column chromatography and their structures were identified by spectral analysis. Of these, compounds **3** and **5** potentially inhibited the catalytic reaction of tyrosinase by blocking the entrance to the active site. Moreover, compounds **3** and **5** altered the turnover rate from substrate to the enzyme product over time. The former underwent simple reversible slow binding with the enzyme, while the latter initially formed a complex (EI) with the enzyme quickly, followed by an isomerization step to create a new enzyme-type complex (E*I). Phlorotannins containing flexible phloroglucinol to eckol backbone were found the facts bound easily into active site of tyrosinase. The findings suggested the phloroglucinol may be key moiety for induced fit of inhibitor with enzyme flexibility. The molecular simulation study indicated that the two inhibitors were docked at the active site of tyrosinase with AutoDock scores or −6.59 and −6.68 kcal/mol, respectively. Both compounds used the amino acids His85 and Asn260 to maintain their connection with tyrosinase. These findings imply that 2-phloroeckol (**3**) and 2-*O*-(2,4,6-trihydroxyphenyl)-6,6′-bieckol (**5**) might inhibit the production of melanin products.

## Figures and Tables

**Figure 1 marinedrugs-17-00359-f001:**
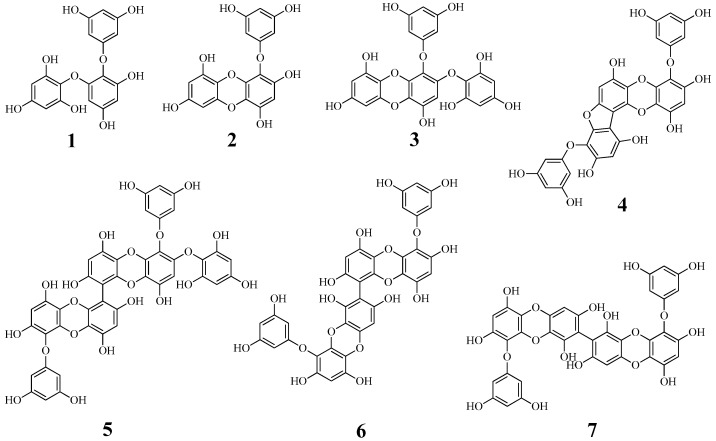
Structure of compounds **1**–**7** from *E. cava*.

**Figure 2 marinedrugs-17-00359-f002:**
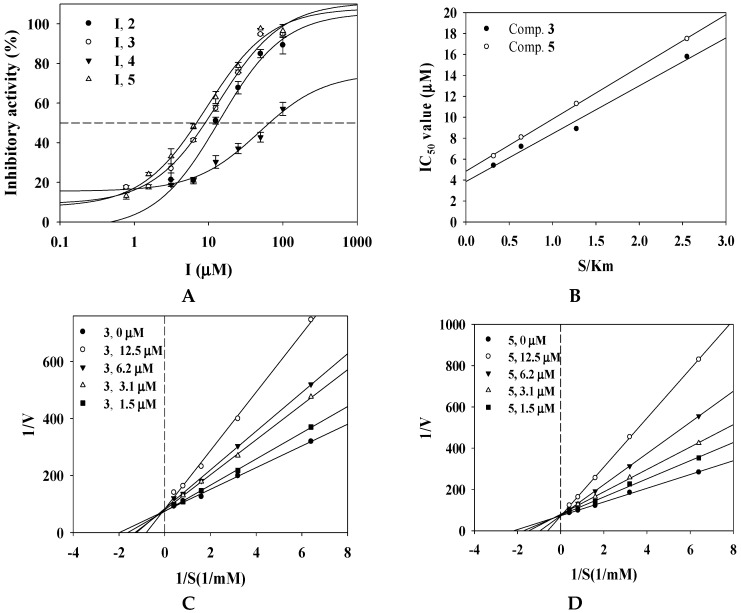
Inhibitory activity of compounds on tyrosinase (**A**). Effects of S/Km ratio on the IC_50_ values (**B**). Lineweaver–Burk (**C**,**D**) and Dixon (**E**,**F**) plots of tyrosinase inhibition by compounds, respectively.

**Figure 3 marinedrugs-17-00359-f003:**
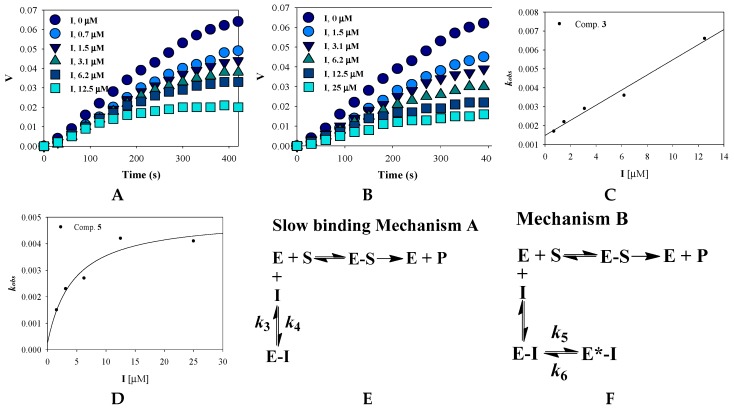
Progress curves for slow-binding inhibition (**A**,**B**) and dependence of the values of *k*_obs_ on the concentration of compounds **3** and **5** (**C**,**D**) of tyrosinase. Mechanism of slow binding inhibitor (**E**,**F**).

**Figure 4 marinedrugs-17-00359-f004:**
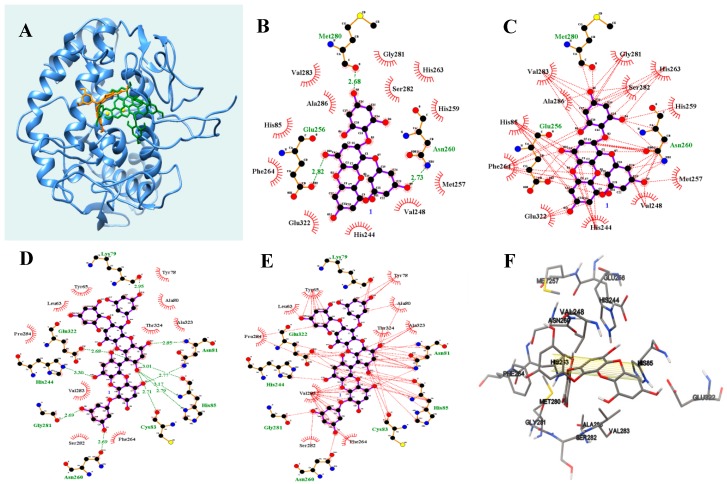
The binding pose (**A**) of two compounds **3** and **5** into tyrosinase (orange: **3**, green: **5**). Hydrogen bonds (**B**,**C**), hydrophobic interactions (**D**,**E**), and 9.4 Å π-π interaction (**F**) of inhibitor with the enzyme.

**Table 1 marinedrugs-17-00359-t001:** Tyrosinase inhibitory activities of compounds **1**–**7**.

	Inhibitory Activity of Compounds on Tyrosinase ^a^		
	100 μM (%)	IC_50_ (μM)	Binding Mode (*K_i_*, μM)
**1**	26.7 ± 1.0	N.T. ^c^	N.T. ^c^
**2**	86.0 ± 4.7	13.5 ± 0.1	Reported as non-competitive ^d^
**3**	94.7 ± 1.4	7.0 ± 0.2	Competitive (8.2 ± 1.1)
**4**	57.1 ± 3.3	66.4 ± 0.1	N.T. ^c^
**5**	86.9 ± 0.9	8.8 ± 0.1	Competitive (5.8 ± 0.8)
**6**	41.2 ± 4.3	N.T. ^c^	N.T. ^c^
**7**	43.3 ± 1.5	N.T. ^c^	N.T. ^c^
Kojic acid ^b^		25.0 ± 0.4	

^a^ All compounds examined in a set of triplicated experiment; ^b^ Positive control; ^c^ Not tested; ^d^ Reference [12].

**Table 2 marinedrugs-17-00359-t002:** Kinetics parameters of tyrosinase by compounds **3** and **5**.

Compound	*K*_3_ (mMs^−1^)	*k*_4_ (s^−1^)	*K*_5_ (s^−1^)	*K_6_* (s^−1^)	*k*app *i* (μM)
**3**	0.0002	0.0013	-	-	6.5 μM
**5**	-	-	0.0047	0.0003	4.4 μM

**Table 3 marinedrugs-17-00359-t003:** Hydrogen bonding interactions and Autodock score between tyrosinase and inhibitors

Compound	Hydrogen Bonds (Å)	Binding Energy (kcal/mol)
**3**	Glu256(2.82), Asn260(2.73), Met280(2.68)	−6.59
**5**	Lys79(2.95), Asn81(2.85), 2.77), Cys83(2.71), His85(2.77, 3.17, 2.79), His244(3.30), Asn260(2.69), Gly281(2.69), Glu322(2.68)	−6.68

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
