# Peer review of "Slow-Binding Inhibition of Tyrosinase by *Ecklonia cava* Phlorotannins"

_marinedrugs, 2019, doi:10.3390/md17060359_

Reviewer 1 Report

In the manuscript submitted to Marine Drugs (code 526877) authors works on the slow-binding inhibition of tyrosinase by Ecklonia cava phlorotannins. This reviewer suggest the publication in Marine Drugs after minor revision.

Theme, techniques and the application part.is interesting and adequated. Results could be interesting for the readers of Marine Drugs.

Minor comments:
* In Abstract, line 11: The inhibitory activity of seven phlorotannins (1–7)? What means? Please avoid the use of abreviations in an abstract.
* Please improve the abstract, introducing more information.

Author Response

Covering Letter to Reviewers’ Comments on Original Manuscript

Manuscript ID: marinedrugs-526877

Title: Slow-binding inhibition of tyrosinase by Ecklonia cava phlorotannins

Co-corresponding Authors: Prof. Young Ho Kim and Dr. Seo Young Yang

Authors: Jang Hoon Kim, Sunggun Lee, Saerom Park, Ji soo Park

The authors would like to thank the Editor for giving us the opportunity to revise this manuscript by addressing the reviewers’ concerns. We also appreciate two reviewers’ insightful comments and suggestions. Below, we addressed the reviewer’s comments and suggestion, point-by-point. We gave revision in the main text accordingly and highlighted by yellow color. The original reviewer’s comments are written by black color, with our response to the reviewers’ comments interspersed by blue color followings:

Reviewers' comments
-Reviewer 1

Comments and Suggestions for Authors

In the manuscript submitted to Marine Drugs (code 526877) authors works on the slow-binding inhibition of tyrosinase by Ecklonia cava phlorotannins. This reviewer suggest the publication in Marine Drugs after minor revision.

Theme, techniques and the application part.is interesting and adequated. Results could be interesting for the readers of Marine Drugs.

Minor comments:
* In Abstract, line 11: The inhibitory activity of seven phlorotannins (1–7)? What means? Please avoid the use of abreviations in an abstract.
* Please improve the abstract, introducing more information.

Answer: Seven phlorotannis (1-7) isn’t abbrevaiation. However, we added in their name “triphlorethol A (1), eckol (2), 2-phloroeckol (3), phlorofucofuroeckol A (4), 2-O-(2,4,6-trihydroxyphenyl)-6,6′-bieckol (5), 6,8′-bieckol (6), and 8,8′-bieckol (7)” according to reviewer’s comment in abstract.

Reviewer 2 Report

General comment:

The manuscript provides interesting results about the inhibitory effects of phlorotannins isolated from E. cava, as well as valuable data about their inhibition mechanism.

The paper could be recommended for publication after some revisions. The comments and questions are below:

Abstract:

Line 11 - Ecklonia cava should be in italic.

A brief introduction as well as the purpose of this study should be given in the abstract.

Introduction:

Line 41 – E. cava should be in italic format

Line 43 – where it is written “exhibited competitive inhibitory and slow-binding inhibition of tyrosinase” should be “exhibited competitive and slow-binding inhibition of tyrosinase”

Results and discussion:

The isolation and identification section is very shallow. Authors should present RMN data at least as an appendix and briefly discuss it in the main text.

Authors should consolidate with previous data in literature. What did other authors describe? Are the results in line with previous works? Have such effects ever been described for compounds 3 and 5? The whole results and discussion section is seriously lacking of the discussion and comparison with other studies.

Line 116 – add a space between “ofcompound”

Methods:

Line 180 – Singular and plural misunderstanding. Where it reads “they was”, should read “they were” or “it was”

Conclusions:

Line 185 – “E. cava” should be in italic

The conclusions section seems like a summary of the results. Authors should conclude with an interpretation of the key results, highlighting the novelty and relevance of the work.

References:

There are several references lacking of spaces between words or having extra spaces: Ref: 1; 2; 4; 5; 9 and 11

Several refs have issue numbers, others do not. Should be uniformized according to the journal references guidelines.

Attention to ref 17.

Author Response

Covering Letter to Reviewers’ Comments on Original Manuscript

Manuscript ID: marinedrugs-526877

Title: Slow-binding inhibition of tyrosinase by Ecklonia cava phlorotannins

Co-corresponding Authors: Prof. Young Ho Kim and Dr. Seo Young Yang

Authors: Jang Hoon Kim, Sunggun Lee, Saerom Park, Ji soo Park

The authors would like to thank the Editor for giving us the opportunity to revise this manuscript by addressing the reviewers’ concerns. We also appreciate two reviewers’ insightful comments and suggestions. Below, we addressed the reviewer’s comments and suggestion, point-by-point. We gave revision in the main text accordingly and highlighted by yellow color. The original reviewer’s comments are written by black color, with our response to the reviewers’ comments interspersed by blue color followings:

Reviewers' comments

-Reviewer 2

Comments and Suggestions for Authors

General comment:

The manuscript provides interesting results about the inhibitory effects of phlorotannins isolated from E. cava, as well as valuable data about their inhibition mechanism.

The paper could be recommended for publication after some revisions. The comments and questions are below:

Abstract:

Line 11 - Ecklonia cava should be in italic.

A brief introduction as well as the purpose of this study should be given in the abstract.

Answer: We revised “Ecklonia cava” to “Ecklonia cava”.

We added in “Tyrosinase inhibitors improve skin whitening by inhibiting the formation of melanin precursors in the skin” in abstract.

Introduction:

Line 41 – E. cava should be in italic format

Answer: We revised “E. cava” to “E. cava”.

Line 43 – where it is written “exhibited competitive inhibitory and slow-binding inhibition of tyrosinase” should be “exhibited competitive and slow-binding inhibition of tyrosinase”

 Answer: We deleted “inhibitory”.

Results and discussion:

The isolation and identification section is very shallow. Authors should present RMN data at least as an appendix and briefly discuss it in the main text.

Answer: We appreciate the reviewer’s comment. However, these compounds were known in previous reports. We added NMR and Mass signals based on supporting data in extraction and isolation section.

Authors should consolidate with previous data in literature. What did other authors describe? Are the results in line with previous works? Have such effects ever been described for compounds 3 and 5? The whole results and discussion section is seriously lacking of the discussion and comparison with other studies.

Answer: We write the additional discussion on our results for solving lacking parts by reviwer’s comments. Especially, compounds 3 and 5 reported few bioactivities. Fortunately, two phlorotannins showed the inhibitory activity within 10 micromole concentration on tyrosinase.

Line 116 – add a space between “ofcompound”

 Answer: We revised “ofcompound” to of “compound”.

Methods:

Line 180 – Singular and plural misunderstanding. Where it reads “they was”, should read “they were” or “it was”

 Answer: We revised as They were.

Conclusions:

Line 185 – “E. cava” should be in italic

Answer: We revised “E. cava” to “E. cava”.

The conclusions section seems like a summary of the results. Authors should conclude with an interpretation of the key results, highlighting the novelty and relevance of the work.

Answer: Compounds 3 and 5 have been reported few component and bioactivity studies. These compounds had highly potential inhibitory activity on tyrosinase. We wrote the key of our results in conclusion.  

References:

There are several references lacking of spaces between words or having extra spaces: Ref: 1; 2; 4; 5; 9 and 11

Several refs have issue numbers, others do not. Should be uniformized according to the journal references guidelines.

Attention to ref 17.

Answer: We confirmed our mistake and revised reference according to reviewer’s comments.

Round  2

Reviewer 2 Report

The results seem better discussed now, and the addition of NMR and Mass signals make the isolation and identification section more elucidative.

I found some grammar mistakes, mainly some confusion between singular and plural forms. For example:

Line 67 - Authors are talking about 2 compounds i.e. in plural, so it should be "inhibitors" instead of "inhibitor"

Line 81 - where it reads "plant" and "inhibitor" should read "plants" and "inhibitors"

Line 81 - where it reads "inhibitor" should read "inhibitors"

Line 85 - Where it reads "inhibitor" should read "inhibitors"

Also:

Line 52 - where it reads "catylatic" should read "catalytic"

Therefore, although I'm not a native english speaker and do not feel highly qualified to judge the english writing, I would recommend some english revision

I would only point out some minor mistakes in the references:

Ref 11 - "brown algae not italic" please correct this mistake

Ref 14 - "Ecklonia cava" should be in italic

Author Response

Covering Letter to Reviewers’ Comments on Original Manuscript

Manuscript ID: marinedrugs-526877

Title: Slow-binding inhibition of tyrosinase by Ecklonia cava phlorotannins

Co-corresponding Authors: Prof. Young Ho Kim and Dr. Seo Young Yang

Authors: Jang Hoon Kim, Sunggun Lee, Saerom Park, Ji soo Park

The authors would like to thank the Editor for giving us the opportunity to revise this manuscript by addressing the reviewers’ concerns. Below, we addressed the reviewer’s comments and suggestion, point-by-point. We gave revision in the main text accordingly and highlighted by yellow color. The original reviewer’s comments are written by black color, with our response to the reviewers’ comments interspersed by blue color followings:

Comments and Suggestions for Authors

The results seem better discussed now, and the addition of NMR and Mass signals make the isolation and identification section more elucidative.

I found some grammar mistakes, mainly some confusion between singular and plural forms. For example:

Line 67 - Authors are talking about 2 compounds i.e. in plural, so it should be "inhibitors" instead of "inhibitor"

Answer: We revised “inhibitor” as “inhibitors”.

Line 81 - where it reads "plant" and "inhibitor" should read "plants" and "inhibitors"

Answer: We revised these words according careful reviewer’s comment.

Line 81 - where it reads "inhibitor" should read "inhibitors"

Answer: We revised “inhibitor” as “inhibitors”.

Line 85 - Where it reads "inhibitor" should read "inhibitors"

Answer: In Line 85, we revised “inhibitor” as “inhibitors.

Also:

Line 52 - where it reads "catylatic" should read "catalytic"

Answer: We revised “catylatic” as “catalytic”.

Therefore, although I'm not a native english speaker and do not feel highly qualified to judge the english writing, I would recommend some english revision

Answer:
The English in this document has been checked by at least two professional editors, both native speakers of English. For a certificate, please see:

http://www.textcheck.com/certificate/8ShhV9

We rewrite the added sentence.

I would only point out some minor mistakes in the references:

Ref 11 - "brown algae not italic" please correct this mistake

Ref 14 - "Ecklonia cava" should be in italic

Answer: We revised Ref 11 and 12.